# *Dentex dentex* Frauds: Establishment of a New DNA Barcoding Marker

**DOI:** 10.3390/foods10030580

**Published:** 2021-03-10

**Authors:** Marina Ceruso, Celestina Mascolo, Pasquale De Luca, Iolanda Venuti, Elio Biffali, Rosa Luisa Ambrosio, Giorgio Smaldone, Paolo Sordino, Tiziana Pepe

**Affiliations:** 1Department of Veterinary Medicine and Animal Production, University of Naples Federico II, via F. Delpino, n.1, 80137 Naples, Italy; marina.ceruso@unina.it (M.C.); celeste.mascolo@gmail.com (C.M.); iolandavenuti@gmail.com (I.V.); rosaluisa.ambrosio@unina.it (R.L.A.); 2Department of Research Infrastructures for Marine Biological Resources, Stazione Zoologica Anton Dohrn, Villa Comunale, 80121 Naples, Italy; p.deluca@szn.it (P.D.L.); elio@szn.it (E.B.); 3Department Agricultural Sciences, University of Naples Federico II, via Università, n.100, Portici, 80055 Naples, Italy; giorgio.smaldone@unina.it; 4Department of Biology and Evolution of Marine Organisms, Stazione Zoologica Anton Dohrn, Sicily Marine Centre, via Consolare Pompea, 29, Villaggio Pace, 98167 Messina, Italy; sordino@szn.it

**Keywords:** fish species authentication, mtDNA, mitogenomics, *Dentex dentex*, *Sparidae*

## Abstract

The common dentex (*Dentex dentex* (Linnaeus, 1758)) is an iconic fish in the Mediterranean diet. Due to its commercial and organoleptic importance, this sparid is highly appreciated in European markets and is often subjected to species substitution frauds. Comparative mitogenomics is a suitable approach for identifying new and effective barcode markers. This study aimed to find a molecular tag useful for unequivocally discriminating the sparid species *D. dentex*. The comparison of the complete mitochondrial DNA (mtDNA) sequences of 16 sparid species allowed us to highlight the potential of the *NAD2* gene for direct identification purposes. Common dentex-specific primers were created and successfully evaluated by end-point and real-rime PCR (Polymerase Chain Reaction) for several fish species, achieving amplification only in the *D. dentex*. The method proposed in this study appears fast, simple, and inexpensive and requires affordable instrumentation. This approach provides unambiguous results for the common dentex authentication without the sequencing step. The presence/absence assay for *D. dentex* can be executed in a few hours of lab work. Therefore, national authorities responsible for food safety and traceability could apply and make full use of DNA-testing methods for deterring operators from false seafood declarations.

## 1. Introduction

Fish and fish products have a central role in human nutrition as they embody a source of nutrients and micronutrients with essential health benefits [1]. The family *Sparidae* comprises 38 genera and 159 species [2], several of which are greatly appreciated as seafood (D.M. MIPAAF, 22 September 2017). Some members of the family are commercially valuable for the nutritional properties and organoleptic features of their flesh. *Sparidae* species have external characteristics quite similar to those of species of inferior value belonging to the same or different genera and families [3,4]. Thus, fraud by substitution of valuable species with others of a lower commercial value is very common. Another reason correlated to frauds is that *Sparidae* whole species identification is not always achievable, even if morphological integrity is conserved [5], and becomes harder to realize after processing procedures, when typical external features are lost. Moreover, the demand in the European market of precious fish species is increasing, so they are commonly replaced with species of minor value, raw and processed, imported from non-EU countries from around the Atlantic and Indo-Pacific Oceans.

Among the *Sparidae* family, the common dentex (*Dentex dentex* (Linnaeus, 1758)) is one of the most expensive sparid species, very appreciated in European markets and consequently more often subjected to species substitution frauds [6]. The replacement often occurs with a host of sparids, such as *Dentex gibbosus*, *Cheimerius nufar*, *Dentex barnardi*, *Dentex canariensis*, and *Pagrus pagrus* [6,7,8,9,10,11,12,13,14]. The mitochondrial DNA (mtDNA) sequences currently and indiscriminately used for the identification of all fish species in prepared and processed products belong to the cytochrome b-*Cytb*, cytochrome c oxidase I-*COI*, *16S*, and *12S* genes. Nevertheless, it is increasingly clear that this classical approach, based on standard mtDNA markers, correctly works for some fish species but is less discriminating for others [15,16]. This especially occurs when the nucleotide similarity degree between species is particularly high, as happens in the *Sparidae* family. 

Our purpose was to identify a gene or gene fragment more specific for *Dentex dentex* and with a higher capacity of species differentiation. Therefore, we compared 16 sparid mitogenomes [17,18], 6 of which [19,20,21,22,23,24] were sequenced in our laboratories using a new and effective extraction protocol [25]. Our previous studies in this field led to the identification of two new barcode markers for the *Sparidae* family. The first one is a fragment of the *NAD5* gene, achievable using primers that produce a 505 *bp* amplicon in all sparid species. Like for all standard barcode markers, *NAD5*-based barcoding requires PCR (Polymerase Chain Reaction) amplification and Sanger sequencing for species detection, thereby entailing time-consuming laboratory steps not easy for routine checks [4,17]. The second marker is based on the *NAD2* gene, which is characterized by a high level of divergence in the nucleotide sequence among sparids. This feature is scattered throughout the gene, allowing the design of species-specific primers. This approach was used to study the Sparidae family and allowed the designing of specific *Pagellus erythrinus* primers [18,26]. Therefore, in this research, we focalized on the direct identification of *D. dentex*, a species that for its high commercial and organoleptic value requires an urgent and efficient solution related to species identification against frauds. We applied the strategy based on the study of *NAD2* gene, which thanks to its nucleotide sequence diversity among *Sparidae* allowed the designing of a specific PCR primer to be used for common dentex identification, with no subsequent sequencing.

## 2. Materials and Methods

### 2.1. Sparidae mtDNA Genome Data and Fish Samples

Sixteen complete *Sparidae* mitogenome sequences were analyzed in this study, as reported in Ceruso et al. [21]. Eleven specimens of *D. dentex* from Food and Agriculture Organization (FAO) area 37 were sampled (Table 1 and Figure 1). The geographical origin of *D. dentex* samples was provided by fishing companies, as established by EC Reg. 1224/2009. FAO area 37 was selected since it holds most of the common dentex distribution (www.aquamaps.org (accessed on 26 January 2021)). The sampling spots were classified following the geographical sub-area (GSA) division from the General Fisheries Commission for the Mediterranean (GFCM) (Resolution GFCM/31/2007/2). The common dentex samples were used to assess the PCR primer species specificity, and evaluation was performed on 26 other fish species (Table 2). The fish species other than *Dentex dentex* were carefully chosen with the aim to include (i) those used to substitute *Dentex dentex* (e.g., *Dentex gibbosus*), (ii) phylogenetically related sparid species (e.g., *Pagellus erythrinus*) [18], and (iii) other species commonly found in Mediterranean fish markets. Ten specimens of *Pagellus erythrinus*, the most genetically close to *Dentex dentex* among sparids [21], were collected.

### 2.2. Total Genomic DNA Extraction

Total genomic DNA (gDNA) was extracted from the skeletal muscle using the DNeasy Blood and Tissue Kit (Qiagen, Hilden, Germany) according to the manufacturer’s instructions [18]. The extracted gDNA was quantified using Nanodrop (Thermo Fisher Scientific, Waltham, MA, USA). The extracted DNA had a concentration of 40 ng/µL, while the purity range was in the ratio of 1.8–2.0 at A260/A280. Electrophoretic analysis in 1% agarose gel was performed to verify DNA integrity.

### 2.3. mtDNA Comparative Analysis and End-Point PCR

The whole mtDNA of the 16 sparids was analyzed using a comparative approach based on several bioinformatics tools with the aim to identify the most efficient gene for the common dentex fingerprinting. MtDNA alignment, Hamming distance analysis (HDA), overall mean *p*-genetic distance, nucleotide sequence variability, pairwise and multiple alignments on DNA sequences, and protein-coding gene (PCG) sequences were determined as reported in Ceruso et al. [18]. *NAD2* primers were designed as previously described [27].

The melting temperature (Tm), the secondary structure, self-annealing, and inter-primer binding were verified using Multiple Primer Analyzer (Thermo Fisher Scientific, Waltham, MA, USA). *NAD2* primer’s performances for sparids barcoding were evaluated in silico using Unipro UGENE software [28]. PCR amplification was performed on total gDNA from 11 fresh specimens of *D. dentex* species (Figure 1 and Table 1) and 26 commercially important fish species from *Sparidae* and other teleost families (Table 2).

PCR conditions are reported in Ceruso et al. [18]; the annealing temperature was 63 °C (249 bp) for 30 seconds. PCR products were electrophoresed on 1.5% agarose gel and visualized via ultraviolet transillumination. Amplicons were purified using the QIAquick PCR Purification Kit (Qiagen) and sequenced and analyzed as described in Hall [27]. The achieved sequences were analyzed through a BLAST analysis on GenBank to identify the species and evaluate the concordance between morphological and molecular analyses [29].

### 2.4. Real-Time PCR 

The DNA extracted was employed as a template for real-time/quantitative PCR (qPCR) analysis, performed in a Viia7 real-time PCR system (Applied Biosystems) at 1:10 dilution, using the same primer set as that used in end-point PCR. The volume of each PCR sample was 10 μL, with 5 μL of 2× qPCR SYBR Green Master Mix (Thermo Fisher); 0.01, 0.05, or 0.1 pmol/μL of each primer; and 1 μL of the diluted DNA template (4 ng). All the evaluations were performed in triplicate. Results were obtained using 0.01 pmol/µL of each primer. The graphic was obtained using Excel. 

## 3. Results

### 3.1. Sparidae mtDNA Analysis 

The HDA revealed that the genetic dissimilarity among the mtDNA of *Dentex dentex* and other sparids is in a range of 6% (*D. gibbosus*) and 14% (*Achantopagrus schlegelii*, *Rhabdosargus sarba*, *Sparus aurata*). This result suggests that the common dentex is very similar to the other sparid species. In agreement with previous work [30,31,32], the HDA indicated *D. dentex* and *D. gibbosus* as the phylogenetically closest pair of sparid species, with 6% of genetic dissimilarity. 

The HDA was performed gene by gene between *D. dentex* and other sparid species [17]. Ranges of genetic dissimilarity were the following: *ATP6* (11–25%), *ATP8* (7–24%), *COI* (8–18%), *COII* (6–16%), *COIII* (8–19%), *Cytb* (8–19%), *NAD1* (11–22%), *NAD2* (12–26%), *NAD3* (11–23%), *NAD4* (9–23%), *NAD4l* (8–20%), *NAD5* (10–22%), and *NAD6* (10–24%). These findings suggest that the *NAD2* gene has a higher dissimilarity value (12%) between the sibling species *D. dentex* and *D. gibbosus* (Figure 2).

Therefore, the *p*-genetic distance evaluation among all mtDNA displayed that the *NAD* group genes have a higher sequence distance than *COI* and *Cytb* genes (*ATP6* 0.24, *ATP8* 0.18, *COI* 0.16, *COII* 0.16, *COIII* 0.16, *Cytb* 0.19, *NAD1* 0.21, *NAD2* 0.25, *NAD3* 0.2, *NAD4* 0.24, *NAD4l* 0.2, *NAD5* 0.24, *NAD6* 0.23). Thus, the highest divergence values were obtained for *NAD2*, *NAD4*, and *NAD5*.

Further, more significative values of nucleotide sequence variability were detected in the *NAD* gene group (*NAD1* 39%, *NAD2* 50%, *NAD3* 39%, *NAD4L* 39%, *NAD4* 43%, *NAD5* 41%, *NAD6* 44%). These results, in accordance with the previous study on *P. erythrinus* identification [18], confirmed and further highlighted the suitability of the *NAD2* gene fragment for the study of *D. dentex* barcoding.

### 3.2. NAD2 Amplification and Analysis

As suggested by sparid mitogenome analysis, primers for amplifying species-specific nucleotide sequences were designed on a fragment of the *NAD2* gene. Table 3 reports the primer sequences. As described in the Methods section, the primers’ specificity against all *Sparidae* species was first tested in silico. Then, PCR results confirmed that the primer set allows amplification only on *D. dentex* mtDNA. PCR amplification was obtained in all geographical specimens of *D. dentex* used in this study, while no amplification occurred when using mtDNA of the other tested species (Figure 3).

Given the high intra-sparid variability of *NAD2*, we analyzed intraspecific variation to verify the reliability of this gene sequence in species control. All *D. dentex NAD2* amplicons were sequenced. Sequencing of the *NAD2* fragment in all geographically disjunct specimens of *D. dentex* analyzed in this study showed an intraspecific genetic variation, ranging from 0% and 0.7%, with two different nucleotides found in 2 out of 11 specimens (DD1 and DD2), further supporting *NAD2* reliability for species characterization. Sequence comparison with databases confirmed the correct species identification, with a range of similarity scores between 98% and 100%.

### 3.3. Real-Time PCR

The real-time PCR approach is very useful because the instrument measures the presence of an amplification product during the reaction so that the operator does not need the electrophoresis step to read the result. The reaction produces two pieces of information, the Ct (the threshold cycle: the first reaction cycle at which the product is well detectable) and the Tm (the melting temperature of the amplification product, which is highly specific for each DNA fragment). We tested our PCR amplification in a real-time PCR to verify if this approach is suitable for *D. dentex* identification. We observed a specific amplification in all the *D. dentex* specimens analyzed (a Ct between 28 and 34 and a Tm of the products around 80 °C, as expected), while all the other species tested provided an undetectable signal, indicated by a Ct of 40, due to a very low amount of unspecific product, with a Tm different from the expected one (Figure 4).

## 4. Discussion 

In 2016, the total catch of the common dentex was around 1397.11 tons [33], with 1.353 tons from fishing and 44.11 tons from aquaculture. Most of the fish (90% of world production) comes from the Mediterranean Sea, from countries such as Greece, Libya, Italy, Tunisia, and Spain. Farmed fishes originate only from two countries: Turkey, with a production of 43 tons, and Croatia, with a production of 1.11 tons. The current prices of *Dentex dentex* current range from 15 €/kg for a whole specimen to 60 €/kg for fillets and sliced fish [34]. These numbers only give a little idea of the commercial importance of this species, internationally recognized as valuable and expensive. The use of the common dentex is very widespread, even in luxury restaurants. The inclusion of this species in a menu allows a significant increase in the meal price. 

With all these premises, it is clear how species substitutions of the common dentex are of enormous significance from a commercial point of view. Substitution frauds on *D. dentex*, however, have also nutritional and legal consequences. 

As regards nutritional features, fish species have high variability in the basic chemical and mineral contents. In particular, the proportion of omega-3 fatty acids was twice as high in dentex as in seabream, seabass, or turbot and the proportions of eicosapentaenoic acid (EPA) and docosahexaenoic acid (DHA) were two to four times higher in *D. dentex* [35].

As regards legal aspects, the Official Controls Regulation (EU) 2017/625 gives growing importance to consumer protection and safety against fraudulent practices along the entire agri-food chain. Besides, EU Commission Regulation 1379/2013 (European Commission, 2013) on the common organization of the markets in fishery and aquaculture products has established detailed rules for consumer information to be included on labels and fish species must always be declared. According to Italian legislation, the only species that can be defined as Dentice, avoiding species declaration, is *Dentex dentex*. All other species of the same genus must be entirely specified. For example, the Gibbous snapper (*Dentex gibbosus*) must be declared as Dentice gibboso and the pink snapper (*Cheimerius nufar*) as Dentice indiano to prevent interpretation mistakes (D.M. MIPAAF, 22 September 2017).

In whole specimens, it is possible to differentiate *D. dentex* from the other species belonging to the same genera because it has caniniform teeth that are quite linear and quite high compared to the cogeneric *D. barnardi* and *D. canariensis*, where they are more robust and of a bigger diameter. *Dentex macrophthalmus* is characterized by tiny, protruding, and slightly arched caniniforms, while the teeth of *D. gibbosus* are thinner, even if they are less delicate than those of *D. dentex* [9,36]. This kind of species identification requires very specialized skills. Thus, molecular techniques appear of fundamental importance for these species, in whole and, in particular, in processed products.

In previous research [17,18], the alignment of the complete mtDNAs of 16 *Sparidae* species allowed the identification of two genes, *NAD5* and *NAD2*, whose nucleotide sequences displayed higher interspecific phylogenetic divergence than in standard markers. 

In this study, the *NAD2* gene revealed its suitability as a species-specific and sequencing-free barcoding marker for *D. dentex*. It is important to note that our study allows unequivocal discrimination of *D. dentex* also with the two sparid species, showing the highest degree of mtDNA sequence homology. In particular, the discrimination was possible also with the species *P. erythrinus*, the most phylogenetically related sparid [19], and with *D. gibbosus*, which showed a genetic dissimilarity in the complete mtDNA of only 6% with the common dentex [17]. 

Our proposed DNA-sequencing-free method was able to rapidly and accurately identify and differentiate the *D. dentex* species from its most commonly substituted species in fish markets. The present paper deals with a simple, reliable, and quick method that is useful for routine high-throughput analysis. The designed primers may be used for conventional end-point and real-time PCR methods. The presence/absence assay for *D. dentex* can be executed in a few hours of lab work. This technique can be applied to fresh, frozen, or processed fillets to detect the fraudulent or unintentional misdeclaration of the common dentex. Our research is meant to continue as we design species-specific probe for this and other *Sparidae* species, such as *Pagellus erythrinus*. 

## 5. Conclusions

Mislabeling of fish such as less precious species using names of more valuable ones is an increasing issue in the agri-food production and distribution chain. To improve consumer protection and safety and ensure transparency in the market, competent national authorities responsible for food security and traceability could apply and exploit all available methods, including DNA-based, to discourage operators from false seafood labeling. This research contributes to the molecular traceability of fishery products and to access to clear and comprehensive information for consumers, in agreement with all the European food law policy and in particular with Regulation (EU) 1379/2013.

## Figures and Tables

**Figure 1 foods-10-00580-f001:**
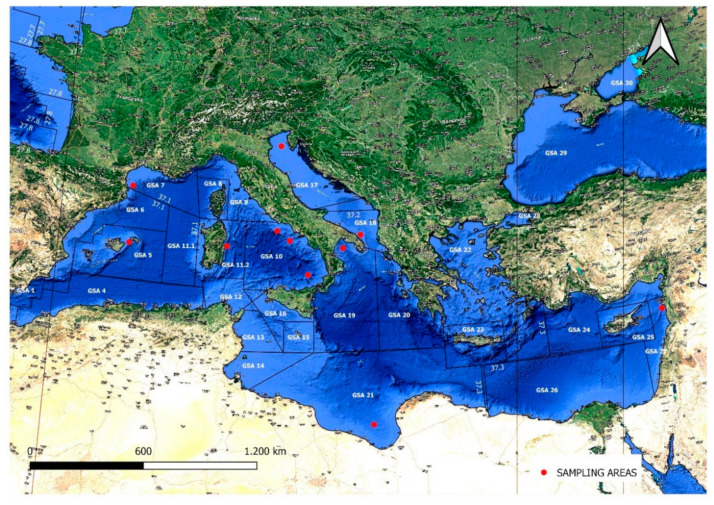
Geographical origins of the *Dentex dentex* specimens (red spots) analyzed in this research. GSA: geographical sub-area.

**Figure 2 foods-10-00580-f002:**
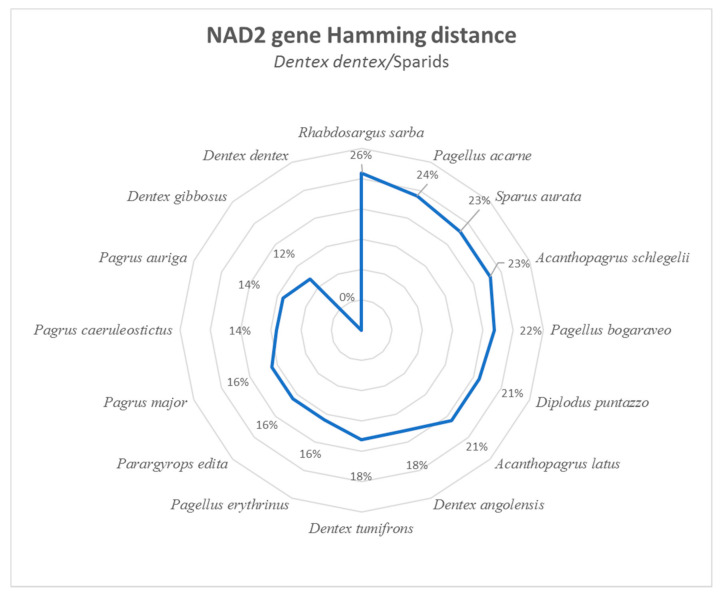
*NAD2* gene Hamming dissimilarity in percent between *D. dentex* and the other *Sparidae* species considered in this study. Starting from *D. dentex*, species are ordered counterclockwise from the less dissimilar to the more distant one.

**Figure 3 foods-10-00580-f003:**
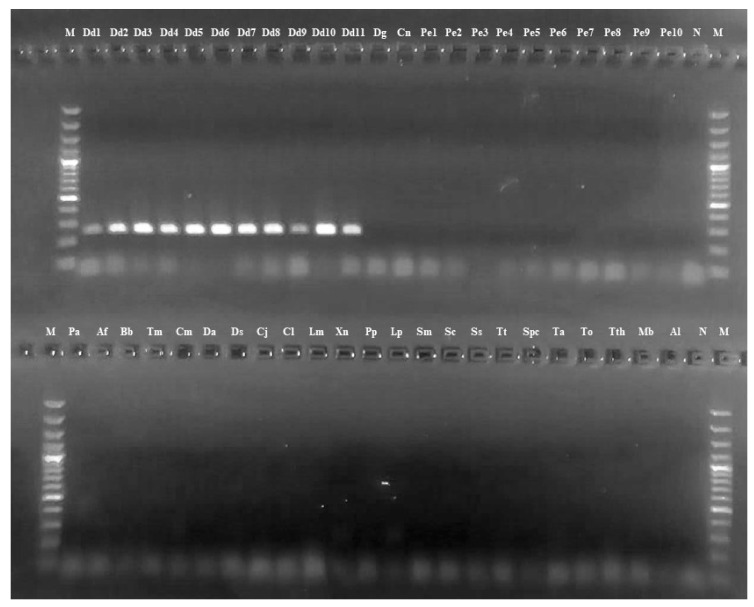
Gel electrophoretic image. End-point PCR amplification of the *NAD2* fragment in the 11 *D. dentex* specimens considered in this study (lanes 2–12, from Dd1 to Dd11). No amplification was obtained for all the other fish species (lanes 13–24 and 25–47). Abbreviations as in Table 1 and Table 2. N: negative control; M: 100 bp ladder.

**Figure 4 foods-10-00580-f004:**
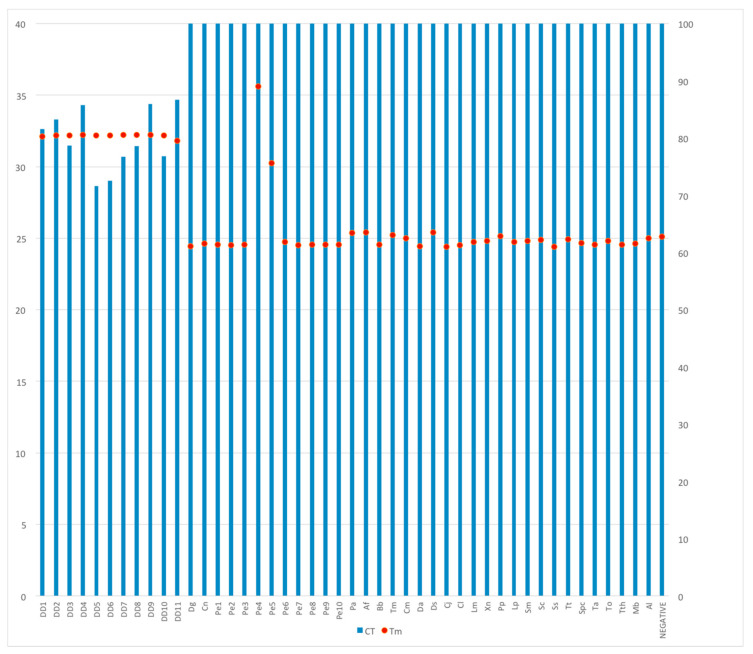
Real-time PCR. Threshold cycles (Ct) on the left and melting temperature (Tm) on the right. Abbreviations as in Table 1 and Table 2.

**Table 1 foods-10-00580-t001:** *D. dentex* samples: geographical sub-areas (GSAs) of the sampling areas and geographic coordinates.

*Dentex dentex*			
Abbreviation	Sampling Area	Latitude	Longitude
Dd1	GSA 5-Balearic Island	39.808662	3.740905
Dd2	GSA 7-Gulf of Lions	42.519035	3.534141
Dd3	GSA 10-South Tyrrhenian	38.771348	14.946960
Dd4	GSA 10-Central Tyrrhenian	40.409446	13.798889
Dd5	GSA 11.2-Sardinia (East)	40.044575	9.841622
Dd6	GSA 17-Northern Adriatic	44.980473	13.137520
Dd7	GSA 18-Southern Adriatic Sea	40.668284	18.290108
Dd8	GSA 19-Western Ionian Sea	40.044587	17.147530
Dd9	GSA 21-Southern Ionian Sea	31.475938	18.685467
Dd10	GSA 27-Levante	35.338496	35.708826
Dd11	GSA 10-Ponza Island, Central Tyrrhenian	40.874620	12.985808

**Table 2 foods-10-00580-t002:** Fish species other than *Dentex dentex* evaluated in this research. The source of the common names was the ASFIS (Aquatic Sciences and Fisheries Information System) List of Species for Fishery Statistics Purposes (http://www.fao.org/fishery/collection/asfis/en (accessed on 26 January 2021)).

N°	Scientific Name	Family	Common Name	Abbreviation
1	*Arnoglossus laterna*	Bothidae	Mediterranean scaldfish	Al
2	*Aulopus filamentosus*	Aulopidae	Royal flagfin	Af
3	*Boops boops*	Sparidae	Bogue	Bb
4	*Cepola macrophthalma*	Cepolidae	Red bandfish	Cm
5	*Cheimerius nufar*	Sparidae	Santer seabream	Cn
6	*Chelidonichthys lucerna*	Triglidae	Tub gurnard	Cl
7	*Coris julis*	Labridae	Rainbow wrasse	Cj
8	*Dentex gibbosus*	Sparidae	Pink dentex	Dg
9	*Diplodus annularis*	Sparidae	Annular seabream	Da
10	*Diplodus sargus*	Sparidae	White seabream	Ds
11	*Lithognathus mormyrus*	Sparidae	Sand steenbras	Lm
12	*Lophius piscatorius*	Lophiidae	Angler (= Monk)	Lp
13	*Mullus barbatus*	Mullidae	Red mullet	Mb
14	*Pagellus acarne*	Sparidae	Axillary seabream	Pa
15	*Pagellus erythrinus*	Sparidae	Common pandora	Pe
16	*Pleuronectes platessa*	Pleuronectidae	European plaice	Pp
17	*Sebastes capensis*	Sebastidae	Cape redfish	Sc
18	*Scophthalmus maximus*	Scophthalmidae	Turbot	Sm
19	*Solea solea*	Soleidae	Common sole	Ss
20	*Spondyliosoma cantharus*	Sparidae	Black seabream	Spc
21	*Thunnus thynnus*	Scombridae	Atlantic bluefin tuna	Tth
22	*Thunnus albacares*	Scombridae	Yellowfin tuna	Ta
23	*Thunnus obesus*	Scombridae	Bigeye tuna	To
24	*Trachurus trachurus*	Carangidae	Atlantic horse mackerel	Tt
25	*Trisopterus minutus*	Gadidae	Poor cod	Tm
26	*Xyrichtys novacula*	Labridae	Pearly razorfish	Xn

**Table 3 foods-10-00580-t003:** Species-specific *NAD2* primers for *Dentex dentex.* Tm (Melting Temperature), CG (Guanine-Cytosine), nt (nucleotide).

N°	Primer Name	5′ → 3′ Sequence	Tm (°C)	CG (%)	nt	A	T	C	G
**1**	Fw108	CACCCTAGCTATCCTCCCCCTCATAGC	72.4	59.3	27	5	6	14	2
	Rev330	AATAACTTCGGGGAGTCACGAGTGTAGG	71.1	50.0	28	8	6	4	10

## Data Availability

Data is contained within the article.

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
