# Peer review of "Dentex dentex Frauds: Establishment of a New DNA Barcoding Marker"

_foods, 2021, doi:10.3390/foods10030580_

Round 1

Reviewer 1 Report

Review of paper Food-1109143

The paper “Dentex dentex frauds: estabishemnte of a new DNA barcoding marker” by Ceruso et al. describes the design of specific markers to detect fraud in fish species. The results obtained with real-time PCR are very interesting. This technique allows quantification, and parameters such as the efficiency of the systems, and the limits of detection (LOD) and limits of quantification (LOQ) must be included. In addition, these results should be contrasted with samples of processed foods.

In this sense, the article presented is of great interest, however, some mayor modification are necessary.

Author Response

Rev1

The paper “Dentex dentex frauds: the establishment of a new DNA barcoding marker” by Ceruso et al. describes the design of specific markers to detect fraud in fish species. The results obtained with real-time PCR are very interesting. This technique allows quantification, and parameters such as the efficiency of the systems, and the limits of detection (LOD) and limits of quantification (LOQ) must be included.

The Real-Time PCR technique is routinely used to evaluate differential expression of genes or absolute count of nucleic acid molecules. We use Real-Time PCR to assess the presence or absence of an amplicon, making it similar to a terminal PCR and gel electrophoresis, with a simple yes/no answer, and this kind of approach does not need to include limits of detection (LOD) and limits of quantification (LOQ) as no variation is detected among the analyzed samples. Results were obtained using 4 ng of DNA template (as now reported in the text). Our goal is to find the simplest possible tool for species identification usable almost anywhere. For the same reason, we do not consider the efficiency of the system from a quantitative point of view. We tested our capability to amplify a single, specific band with different primer dilutions, obtaining each time similar results, so we decided to use 0.01 mM primers, which showed good sensitivity and specificity, for all the subsequent reactions.

In addition, these results should be contrasted with samples of processed foods.

Considering the small amplicon length (249 bp) analyzed in our work on D. dentex, we expect that similar discrimination capacity could be reasonably obtained with samples of processed foods, as reported by related literature (e.g. Armani et al., 2015). Here, we have designed primers for amplifying a small PCR product in order to make the method applicable also to processed products, in which DNA is variably degraded. By the way, it is important to specify that D. dentex-based processed products do not exhibit widespread diffusion in the market.

In this sense, the article presented is of great interest, however, some major modifications are necessary.

Reviewer 2 Report

Interesting work that should be a valuable contribution to mislabeling of fish for consumption.

I have some minor questions/concerns that addressing may help the manuscript:

-I suggest changing reference to RT-PCR to quantitative/qPCR to avoid any confusion with reverse transcriptase PCR (the other RT-PCR). I understand that your method is not necessarily meant to be quantitative, but I still think using 'qPCR' is the more correct term here.

-Related to the PCR, why did you not develop a species-specific probe for the detection of common dentex? It would be more reliable (and potentially quantitative for future work beyond mislabeled fish fillets) than the melting temperatures of PCR amplicons.

-Why did you use Hamming distance for measuring the genetic distances? It seems to have been developed for microarrays and is not commonly applied to DNA sequence comparisons. Something like K2P, corrected p-distances, etc., would be more appropriate. 

-Define FAO when it is first mentioned in the methods.

-Several times you say 'free-sequencing' when you mean to say 'sequencing-free'.

Author Response

Rev2

Interesting work that should be a valuable contribution to mislabeling of fish for consumption.

I have some minor questions/concerns that addressing may help the manuscript:

-I suggest changing reference to RT-PCR to quantitative/qPCR to avoid any confusion with reverse transcriptase PCR (the other RT-PCR). I understand that your method is not necessarily meant to be quantitative, but I still think using 'qPCR' is the more correct term here.

We agree with the reviewer #2’s comment on the use of the term RT-PCR, that we accordingly replaced with qPCR.

-Related to the PCR, why did you not develop a species-specific probe for the detection of common dentex? It would be more reliable (and potentially quantitative for future work beyond mislabeled fish fillets) than the melting temperatures of PCR amplicons.

We thank reviewer #2 for highlighting the usefulness of TaqMan probes for discriminating Dentex dentex, as well as other commercially valuable fishes, from closely related species. At this junction, our aim was to develop a rapid and cheap method based on conventional PCR, of immediate use in every laboratory. On the other hand, the use of TaqMan probes may be more expensive and requires skilled operators. However, as now stated in the main text, discussion section, “our research is meant to continue by designing species-specific probe for this and other sparidae species, such as Pagellus erythrinus”.

-Why did you use Hamming distance for measuring the genetic distances? It seems to have been developed for microarrays and is not commonly applied to DNA sequence comparisons. Something like K2P, corrected p-distances, etc., would be more appropriate. 

A complete analysis of genetic distance among sparidae species, including D. dentex, was already reported in previous papers of the same research group (Ceruso et al, 2019, 2020). In our paper we write that “The whole mtDNA of the sixteen sparids was analyzed using a comparative approach, based on several bioinformatics tools with the aim to identify the most efficient gene for the Common dentex fingerprinting. MtDNAs alignment, Hamming Distance Analysis (HDA), overall mean p-genetic distance, nucleotide sequence variability, pairwise and multiple alignments on DNA sequences, and Protein-Coding Gene (PCG) sequences were determined as reported in Ceruso et al., 2020 [18].” (lines 108-112). The Figure 2 of this paper is just an elaboration of previous results. We mostly focused our attention on the NAD2 gene, for which the p-genetic distance analysis (0.25) and nucleotide sequence variability (50%) confirmed HDA results.  

In addition, we performed K2P analysis for all genes. For your convenience, we report below K2P analysis related to NAD2. Results confirmed all other findings, showing that the NAD2 gene has a higher dissimilarity value (0.13) between the sibling species D. dentex and D. gibbosus (eg. COI 0.083; COII 0.061; CYTB 0.084; 12S 0.029; 16S 0.034).

-Define FAO when it is first mentioned in the methods.

Done

-Several times you say 'free-sequencing' when you mean to say 'sequencing-free'.

Done

References

Ceruso, M.; Mascolo, C.; Anastasio, A.; Pepe, T.; Sordino, P. Frauds and fish species authentication: study of the complete mitochondrial genome of some Sparidae to provide specific barcode markers. Food Control 2019, 103, 36–47.

Ceruso, M.; Mascolo, C.; De Luca, P.; Venuti, I.; Smaldone, G.; Biffali, E.; Anastasio, A.; Pepe, T.; Sordino, P. A Rapid Method for the Identification of Fresh and Processed Pagellus erythrinus Species against Frauds. Foods 2020, 9, 1397.

Round 2

Reviewer 1 Report

The revised text has been improved and can be published in the journal

Reviewer 2 Report

Thanks for addressing my comments.